# Impact of Combined Heat and Salt Stresses on Tomato Plants—Insights into Nutrient Uptake and Redox Homeostasis

**DOI:** 10.3390/antiox11030478

**Published:** 2022-02-28

**Authors:** Bruno Sousa, Francisca Rodrigues, Cristiano Soares, Maria Martins, Manuel Azenha, Teresa Lino-Neto, Conceição Santos, Ana Cunha, Fernanda Fidalgo

**Affiliations:** 1GreenUPorto-Sustainable Agrifood Production Research Centre & INOV4AGRO, Biology Department, Faculty of Sciences, University of Porto, Rua do Campo Alegre s/n, 4169-007 Porto, Portugal; cfsoares@fc.up.pt (C.S.); up201306375@fc.up.pt (M.M.); ffidalgo@fc.up.pt (F.F.); 2Biology Department and CBMA-Centre of Molecular and Environmental Biology, School of Sciences, University of Minho, Campus de Gualtar, 4710-057 Braga, Portugal; tlneto@bio.uminho.pt (T.L.-N.); accunha@bio.uminho.pt (A.C.); 3CIQ-UP, Chemistry and Biochemistry Department, Faculty of Sciences, University of Porto, Rua do Campo Alegre s/n, 4169-007 Porto, Portugal; mazenha@fc.up.pt; 4LAQV/REQUIMTE, Laboratory of Integrative Biology and Biotechnology (IB2), Biology Department, Faculty of Sciences of University of Porto, Rua do Campo Alegre s/n, 4169-007 Porto, Portugal; csantos@fc.up.pt

**Keywords:** oxidative stress, antioxidant system, *Solanum lycopersicum*, climate change, salinity, high temperatures, combined stress

## Abstract

Currently, salinity and heat are two critical threats to crop production and food security which are being aggravated by the global climatic instability. In this scenario, it is imperative to understand plant responses to simultaneous exposure to different stressors and the cross-talk between underlying functional mechanisms. Thus, in this study, the physiological and biochemical responses of tomato plants (*Solanum lycopersicum* L.) to the combination of salinity (100 mM NaCl) and heat (42 °C; 4 h/day) stress were evaluated. After 21 days of co-exposure, the accumulation of Na^+^ in plant tissues was superior when salt-treated plants were also exposed to high temperatures compared to the individual saline treatment, leading to the depletion of other nutrients and a harsher negative effect on plant growth. Despite that, neither oxidative damage nor a major accumulation of reactive oxygen species took place under stress conditions, mostly due to the accumulation of antioxidant (AOX) metabolites alongside the activation of several AOX enzymes. Nonetheless, the plausible allocation of resources towards the defense pathways related to oxidative and osmotic stress, along with severe Na toxicity, heavily compromised the ability of plants to grow properly when the combination of salinity and heat was imposed.

## 1. Introduction

Human societies, for millennia, have been built around stable and efficient agricultural practices meeting a wide range of human needs, most notably food, fibers, fuels, and raw materials. Up until recently, agriculture has been evolving and serving its purpose, but the increasing world population associated with a frightening scenario of climatic instability is taking a heavy toll on the ability of this sector to efficiently respond to the needs of our modern society [1,2]. In fact, the total arable area has been rapidly declining worldwide due to soil degradation (e.g., heavy salinization, nutrient deficiency, contamination) and the higher occurrence of drastic climatic events, such as extreme temperatures, drought, and floods [3,4].

Nowadays, it is estimated that around 4 Mha of European soil is moderately to highly degraded by secondary salinization, mostly due to irrigation with saline water and poor drainage conditions, which is one of the main factors driving the desertification of the Mediterranean coast [5]. Alarmingly, this trend will escalate even more due to the impacts of other climate-change-related events projected to be intensified, such as heat and drought waves. The increases in global temperatures will lead to an increment of irrigation with poor quality water (e.g., saline) to prevent drought-related crop losses, as a consequence of low precipitation and reduced extension of watercourses. Moreover, it will cause higher salt build-ups after evaporation, thus aggravating the soil salinization problem [5,6,7]. The Mediterranean Basin, which, throughout this century, is expected to face up to 50 days per year with maximum daily temperatures above 40 °C (an increase of 10–25 days per year, considering the present scenario), will be particularly affected, resulting in great losses of agricultural productivity [8].

Up to now, there has been extensive literature regarding the effects of salinity or high temperatures on the growth and development of several plants, as both conditions can vastly affect the germination and developmental processes, impair the photosynthetic performance, and compromise water relations and the nutrient balance, ultimately leading to reduced yield and a loss of viability (as reviewed by Wahid et al. [9] and Parihar et al. [10]). Indeed, a proper nutrient supply is of extreme importance for optimal development and growth. However, salinity deeply affects nutrient balance by lowering the assimilation of potassium (K^+^), calcium (Ca^2+^), and magnesium (Mg^2+^), which are of high importance in numerous pathways and networks [10,11], while simultaneously increasing the uptake of sodium (Na^+^) and chloride (Cl^−^), which can be highly toxic and interfere with several essential cellular processes [11]. Nonetheless, and while both stressors may lead to similar end results in plant growth through different affected pathways, one feature that is commonly and similarly affected by the exposure to salt or heat is the cellular redox status. In fact, the stressors may cause an overproduction of reactive oxygen species (ROS) and/or the inhibition of antioxidant (AOX) machinery, ultimately leading to a loss of cell viability and death [12].

However, and despite the knowledge regarding the effects of different abiotic stressors, in a real environmental context, crops are exposed to a multitude of factors whose impacts on the plants’ physiological performance are not always easily extrapolated from what occurs in the presence of an individual stressor [13,14,15,16,17]. Despite that, very few authors have tackled the impacts of a consistently warmer and more saline environment (either through soil salinization or poor water quality) on plant species [18,19,20,21,22].

Thus, and considering all that has been mentioned, more studies must focus on important crops that are seriously threatened by the changing climate. For instance, countries in the Mediterranean region are highly associated with tomato (*Solanum lycopersicum* L.) production—where it has been cultivated for centuries—with Spain and Portugal consistently being in the top five tomato producers in Europe. However, this crop faces serious threats, with it being reported that the forecasted climate change will severely affect tomato yield, with high temperatures and soil salinity being the major stress factors acting in this region [5,8]. Nevertheless, in this species, only two studies have been conducted so far. Rivero et al. [21] and Lopez-Delacalle et al. [20] showed that in comparison to individual treatments, the combination of salt and heat can differentially affect several pathways and improve water efficiency. However, the use of hydroponic growing systems and persistent but lower temperatures (35 °C) than those considered in current projections might not accurately reflect the response of a usually potted and summer-grown plant, especially when these stressors are only applied for a short duration.

In this sense, the main goal of this work is to understand how periodic exposure to high temperatures (42 °C) and irrigation with saline water [100 mM sodium chloride (NaCl)] affects the performance of tomato plants under pot conditions. To address these objectives, several biological questions need to be answered throughout this research: (a) How does the combination of heat and salinity affect the growth and development of tomato plants? (b) Does this combination of stressors disrupt the redox and nutrient balance of these plants? (c) How does the AOX system respond to these stress-induced redox fluctuations? (d) Is the combination of heat and salinity merely the sum of its parts, or are there new and complex mechanisms triggered by this situation that need to be carefully considered?

## 2. Materials and Methods

### 2.1. Plant Material and Growth Conditions

Seeds of *Solanum lycopersicum* L. var. *cerasiforme* (cherry tomato) were surface disinfected by immersion in 70% (*v*/*v*) ethanol for 5 min, followed by a 5-min incubation in 20% (*v*/*v*) commercial bleach (5% active chloride), containing 0.02% (*w*/*v*) Tween^®^-20 (Sigma-Aldrich^®^, Steinheim, Germany). Both procedures were performed under constant agitation, followed by successive clean-ups with deionized water (dH_2_O). Then, seeds were placed in Petri dishes (10 cm diameter) containing solidified (0.675% (*w*/*v*) agar) 0.5× MS medium, including Gamborg B5 vitamins (pH 5.5–6.0, Sigma-Aldrich^®^, Steinheim, Germany) [23], and left to germinate for 7 days in a growth chamber under controlled conditions (16 h light/8 h dark, 25 °C, 150 µmol m^−2^ s^−1^). After this period, plantlets with similar size and development were transferred to plastic pots filled with 600 mL Siro Royal universal substrate (SIRO^©^, Mira, Portugal; physicochemical characteristics in Appendix A) and grown under the same controlled conditions as above. To ensure replicability and avoid competition, three plants were sown per pot. During the first week, plantlets were acclimated to the new conditions, being irrigated only with dH_2_O. A total of 32 pots were prepared.

### 2.2. Experimental Design

After the 7-day acclimation period, pots were randomly divided into four trays (one per experimental condition), each containing eight pots, and plants were grown for the next 21 days under the following treatments:CTL (control)—plants were irrigated with dH_2_O;SALT—plants were irrigated every two days with a 100 mM NaCl solution (60 mL per pot);HEAT—plants were irrigated with dH_2_O and daily exposed to 42 °C for 4 h in a twin growth chamber (temperature scaled up to 42 °C);COMBINED—plants were irrigated every two days with a 100 mM NaCl solution (60 mL per pot) and exposed to 42 °C for 4 h daily in a twin growth chamber (temperature scaled up to 42 °C).

The selection of NaCl concentration was based on previous bibliographic records [24,25,26] and on preliminary assays performed in our laboratory (Appendix A). Moreover, according to Ayers and Westcot [27], the level of salinity applied here (equivalent to 11 dS m^−1^, measured with CDM210 MeterLab electrical conductivity meter) in the irrigation water is just slightly above the tolerance threshold for moderately sensitive species (5 to 10 dS m^−1^ irrigation water electric conductivity), such as tomato. Thus, 100 mM NaCl is an adequate concentration to impose salt stress while maintaining an environmentally relevant experimental design. Regarding the heat stress, this was induced by a daily 4 h exposure to 42 °C, based on the projections already mentioned for the Mediterranean region, and was imposed between the 5th and 9th h of light, mimicking the hottest hours in a field situation.

### 2.3. Plant Harvest and Biometric Analysis

After 21 days of growth, plants were collected, thoroughly washed, and divided into roots and shoots, and the length and fresh weight (fw) of both parts were determined for all plants. Then, part of the plant material from each replicate of all experimental conditions was: (i) left to dry in an oven at 60 °C until reaching stable weight to determine the dry weight (dw) and the water content; (ii) immediately used for the estimation of superoxide anion (O_2_^−^) content; or (iii) frozen and macerated in liquid nitrogen and stored at −80 °C until further use.

Since plant water content was affected by the applied stressors, biochemical parameters were expressed on a dw basis, estimated from the tissues’ water content.

### 2.4. Element Quantification—Na^+^, K^+^, Ca^2+^, and Mg^2+^

For the quantification of inorganic elements (Na^+^, K^+^, Ca^2+^, and Mg^2+^), four dried samples of roots and shoots of tomato plants (each sample comprising three plants) were crushed with an ultracentrifuge mill at 8000 rpm (ZM 200, Retsch), and then, three sub-samples (0.3–0.5 g) were digested in a microwave oven with 4 mL of concentrated nitric acid (HNO_3_) and 2 mL 30% (*w*/*v*) hydrogen peroxide (H_2_O_2_). The digestion proceeded at 800 W for 10 min, followed by 5 min at 1000 W and a cooling period of 15 min. Each clear solution obtained was quantitatively transferred to 50 mL volumetric flasks. The analysis was performed by flame furnace atomic absorption spectroscopy (FAAS), operated at the optical and flame parameters recommended for the instrument used (Thermo Scientific, ICE 3300). Calibration was performed with external standards (in 0.5% HNO_3_) in the following ranges: Na^+^ (0.1–0.8 mg L^−1^), K^+^ (0.2–1.6 mg L^−1^), Ca^2+^ (0.3–2.5 mg L^−1^), and Mg^2+^ (0.075–0.5 mg L^−1^). Results were expressed as mg g^−1^ dw.

### 2.5. Determination of ROS Content—Superoxide Anion (O_2_^.−^) and Hydrogen Peroxide (H_2_O_2_)

The estimation of O_2_^−^ content was performed in fresh samples of roots and shoots by monitoring the nitrite formation from hydroxylamine in the presence of O_2_^−^, in accordance with the protocol described by Sharma et al. [28]. In order to estimate O_2_^−^ levels, a standard curve was prepared using sodium nitrite, and the absorbance (Abs) was read at 530 nm. Results were expressed as µmol g^−1^ dw.

The levels of H_2_O_2_ were determined by the titanium sulfate (TiSO_4_) colorimetric method in accordance with de Sousa et al. [29]. The Abs of the yellowish complex, formed when an acidic solution of titanyl ions is mixed with H_2_O_2_, was read at 410 nm and results were expressed as nmol g^−1^ dw, using 0.28 µM^−1^ cm^−1^ as extinction coefficient (ε).

### 2.6. Estimation of the Lipid Peroxidation (LP) Degree

LP was evaluated in accordance with Heath and Packer [30], based on the determination of malondialdehyde (MDA) content, an end product of this process [12]. Abs was read at 532 and 600 nm, with the latter being subtracted from the first to avoid the effects of non-specific turbidity. MDA content was expressed as nmol g^−1^ dw, using ε = 155 mM^−1^ cm^−1^.

### 2.7. Quantification of Proline, Ascorbate (AsA), and Reduced Glutathione (GSH)

Proline levels were determined via a ninhydrin-based colorimetric assay, first described by Bates et al. [31]. Abs was read at 520 nm, and proline content was estimated using a standard curve, prepared with known proline concentrations. The results were then expressed as mg g^−1^ dw.

Reduced ascorbate (AsA) was quantified through the methodology described by Gillespie et al. [32], based on the AsA-mediated reduction of the ferric ion (Fe^3+^) to ferrous ion (Fe^2+^), which then forms a complex with 2-2′-bipyridyl, measurable at 525 nm. The same method was applied to determine the total AsA content, after which samples were treated with dithiothreitol (DTT) to reduce the oxidized portion of this AOX (dehydroascorbate—DHA). Results were expressed as µmol g^−1^ dw after preparing a standard curve with known AsA concentrations. DHA content was calculated by subtracting the reduced AsA from the total AsA pool.

The quantification of GSH (free and reduced glutathione) was performed in accordance with the protocol optimized by Soares et al. [33], which is based on the Glutathione Assay Kit (CS0260; Sigma-Aldrich^®^, Steinheim, Germany). Here, the complex formed between GSH and 5,5-dithio-bis-(2-nitrobenzoic acid) (DTNB) was measured at 412 nm, and GSH levels were estimated from a calibration curve prepared with known GSH concentrations. Results were expressed as nmol g^−1^ dw.

### 2.8. Quantification of Total Thiols and Non-Protein/Protein-Bound Thiols Ratio

Total thiol quantification was accomplished as described by Zhang et al. [34], using DTNB to determine the concentration of sulfhydryl groups (-SH). Non-protein thiol quantification was performed according to the same method, but with the addition of 10% (*w*/*v*) sulfosalicylic acid to allow for protein precipitation. Both quantifications were done by measuring Abs_412 nm_ (using ε of 13,600 M^−1^ cm^−1^), and the results were expressed as µmol g^−1^ dw. Protein-bound thiols were subsequently calculated by subtracting non-protein thiols from the total thiol content.

### 2.9. Enzymatic Activity-Superoxide Dismutase (SOD; EC 1.15.1.1), Catalase (CAT; EC 1.11.1.6), Ascorbate Peroxidase (APX; EC 1.1.11.1), Glutathione Reductase (GR; EC 1.6.4.2), and Dehydroascorbate Reductase (DHAR; EC 1.8.5.1)

The extraction of the main AOX enzymes was performed, under cold conditions, by an adaptation of the method described by Fidalgo et al. [35]. Here, ≈200 mg of frozen shoot and root samples was mixed with 1.5 mL of an extraction buffer composed of 100 mM potassium phosphate buffer (pH 7.3) and supplemented with 1 mM ethylenediamine tetraacetic acid (EDTA), 8% (*v*/*v*) glycerol, 1 mM phenylmethylsulfonyl fluoride (PMSF), 5 mM AsA, and 2% (*w*/*v*) polyvinylpolypyrrolidone (PVPP). After centrifugation (16,000× *g* for 25 min at 4 °C), the supernatant was collected and used for protein quantification and determination of enzymatic activity. Soluble proteins were estimated using the method described by Bradford [36], using bovine serum albumin as standard.

The activity of SOD was determined via a spectrophotometric assay based on the inhibition of photochemical reduction of nitro blue tetrazolium (NBT) [37]. Here, Abs was recorded at 560 nm, and the results were expressed as units of SOD mg^−1^ protein, with one unit of SOD being defined as the amount of enzyme necessary to cause a 50% inhibition of NBT photoreduction.

CAT and APX activity were estimated spectrophotometrically by monitoring the over-time H_2_O_2_ (ε_240 nm_ = 39.4 mM^−1^ cm^−1^) degradation and AsA (ε_290 nm_ = 0.49 M^−1^ cm^−1^) oxidation, respectively. In both cases, H_2_O_2_ was added to start the reaction, and results were expressed as µmol H_2_O_2_ min^−1^ mg^−1^ protein or nmol AsA min^−1^ mg^−1^ protein. These determinations were performed according to the Aebi [38] and Nakano and Asada [39] methods for CAT and APX activity assessment, respectively, being downscaled for microplates, as optimized by Murshed et al. [40].

In a similar way, GR and DHAR activity were also determined through spectrophotometric enzyme kinetics, downscaling the Foyer and Halliwell [41] and Ma and Cheng [42] methods for UV microplates, respectively, as described by Murshed et al. [40]. For GR, NADPH oxidation was monitored over time at 340 nm after adding oxidized glutathione (GSSG) to the mixture, and results were expressed as nmol NADPH min^−1^ mg^−1^ protein, using 6.22 mM^−1^ cm^−1^ as extinction coefficient. DHAR activity levels were determined by adding DHA to the mixture and following its reduction to AsA at 265 nm. Results were expressed as nmol AsA min^−1^ mg^−1^ protein, considering ε_265 nm_ = 14 mM^−1^ cm^−1^.

### 2.10. Statistical Analyses

Every parameter was assessed using at least three biological replicates—defined here as a mixture of the 3 plants of each pot (*n* ≥ 3)—with at least three technical repetitions per assay. Results were expressed as mean ± standard error of the mean (SEM). Differences among treatments were assessed by two-way ANOVA (SALT—0 mM and 100 mM NaCl; HEAT—25 °C and 42 °C (4 h d^−1^)) after checking the normality and homogeneity assumptions. When *p* ≤ 0.05, differences between groups were assessed by Tukey’s post-hoc test. When significance was found for the interaction, a correction for the simple main effects was performed. These analyses were carried out using GraphPad Prism version 8.0.2 for Windows (GraphPad Software, San Diego, CA, USA, www.graphpad.com (accessed on 29 December 2021)), and the results of the ANOVAs are detailed in Appendix A.

A principal component analysis (PCA) was performed to assess the similarities between conditions and the major associations between variables that are responsible for the observed similarities/differences. For this, the average values for each evaluated parameter were plotted, and the first two components were used to make biplots. This analysis was carried out using XLSTAT 2021.2.2 [http://www.xlstat.com (Accessed on 29 December 2021), Addinsoft USA, New York, NY, USA].

## 3. Results

### 3.1. Biometric Analysis—Organ Length, Dry Biomass, and Water Content

The individual stress treatments induced similar growth inhibitions, as seen by the significant decrease in organ elongation (17% and 26% in SALT; 24% and 27% in HEAT for roots and shoots, respectively) in relation to the CTL (Figure 1a,d). The exposure to salt or heat stress also led to identical decreases in dry weight when compared to CTL plants (Figure 1b,e), with inhibition values of around 40% and 30% in roots and shoots, respectively. The combination of stressors imposed a more severe negative effect on both the length and dry weight of tomato plant primary organs (decreases of 46% and 77%; 58% and 71% in roots and shoots, respectively), compared with the CTL, as better visualized in Figure 1g, although no significant differences could be found for the interaction between both factors (Appendix A). Concerning water content (Figure 1c,f), no effects were observed in roots, while the treatment with salt, either alone or in combination, led to a significant reduction in water content in the aerial parts of the plants.

### 3.2. Element Quantification—Na^+^, K^+^, Ca^2+^, and Mg^2+^

Plants from both salt treatments (single or combined) presented a severe increase in the levels of Na^+^ in roots (almost 9-fold and 10-fold for SALT and COMBINED, respectively, in relation to the CTL) as well as in shoots, where plants under combined exposure were, once again, more affected (accumulation of around 8-fold) than those under individual salinity stress (accumulation of 5-fold) (Table 1), with statistical significance being attributed to the interaction between HEAT and SALT (Appendix A). On the other hand, even though the heat treatment also resulted in an altered accumulation of Na^+^ (11% increase in roots and a 19% decrease in shoots, in comparison with the CTL), its levels were much lower than those found in SALT and COMBINED. Curiously, the concentration of K^+^ in plants exposed to salt, whether single or in combination with heat, decreased 32–39% in roots and 31–35% in shoots, while heat stress imposed a 14% increment of this element in roots but a decrease in shoots (14%). A different pattern was observed for Ca^2+^, which was accumulated when plants were exposed to heat (14% and 38% in roots and shoots, respectively) and in the shoots of the individual salt treatment (17%), even though it was decreased in the roots (34%). However, upon combination, the stressors led to diminished levels of Ca^2+^ in both organs when compared to the CTL (63% in roots and 12% in shoots). Lastly, levels of Mg^2+^ in heat-stressed plant tissues were either unaltered (roots) or reduced (by 6% in shoots), while salt stress increased the concentration of this element by 25% and 11% in roots and shoots, respectively. The interaction between stressors was significant in both organs (Appendix A), with plants treated simultaneously with salt and heat presenting 6–10% less Mg^2+^ than control plants.

### 3.3. ROS Content

Regarding O_2_^−^ (Figure 2a,d), all plants exhibited similar levels of this ROS in roots, independently of the applied treatment. In shoots, its levels were decreased by 20% when plants were exposed to salt, whilst the combination of both conditions led to a further reduction (52% in comparison with CTL). Concerning H_2_O_2_ (Figure 2b,e), heat stress, either single or combined with salt, resulted in an equal increment of this ROS in roots (63% in relation to CTL). In shoots, however, H_2_O_2_ levels decreased similarly with all treatments (33% in SALT and HEAT and 36% in COMBINED) compared to the CTL.

### 3.4. LP

LP degree, which was estimated by the MDA content, is shown in Figure 2c,f. When plants were exposed to the stresses, LP diminished equally in shoots in relation to the CTL (56%, 52%, and 67% in SALT, HEAT, and COMBINED, respectively). In roots, the simultaneous exposure to the stressors led to significantly lower values (29%) in comparison with the CTL.

### 3.5. Proline, AsA, and GSH

Proline levels were severely affected by salt in shoots (27-fold) and especially affected in roots (59-fold) in relation to the CTL (Table 2 and Table 3). Under the co-exposure scenario, the accumulation of proline was not as pronounced as the single treatment with salt (with 44- and 17-fold changes being noted in shoots and roots, respectively), with the ANOVA results showing significant interaction between SALT and HEAT (Appendix A). Regarding heat treatment alone, no significant differences were found in relation to the CTL in either roots or shoots.

Total AsA (Table 2 and Table 3) was only negatively affected in the shoots of tomato plants by salt, single or combined, where significant decreases of around 30%, compared with the CTL, were recorded (Table 3). Moreover, a 36% and 31% decrease could be found in the DHA content in shoots of these two treatments (SALT and COMBINED, respectively; Table 3). Lastly, and although shoots of every treatment tended to present lower reduced AsA content than the CTL (Table 3), no statistical significance was achieved.

Concerning GSH, the ANOVA results (Appendix A) showed a positive interaction between both treatments. Indeed, its content in roots was only altered upon the simultaneous exposure to salt and heat, being 48% higher than in the CTL (Table 2). On the contrary, this thiol was decreased in the shoots of plants exposed to all treatments (Table 3). However, as can be seen, salinity led to a greater reduction (29%) than those found in HEAT and COMBINED treatments (13% and 16%, respectively).

### 3.6. Thiols

Total thiol content is presented in Table 2 and Table 3. In roots, heat stress led to an increase of around 35% in total thiols regardless of salt co-exposure. In shoots, total thiols were only negatively affected by heat alone (32% compared with the CTL), although a significant interaction was perceived between SALT and HEAT (Appendix A), related to the relatively higher values found in the COMBINED condition. The ratio between non-protein and protein-bound thiols remained unaffected, the exception being the shoots of heat-treated plants (increase of 76% in relation to the untreated plants).

### 3.7. Enzymatic Activity (SOD, CAT, APX, DHAR, and GR)

Results regarding the activity of the AOX enzymes are presented in Figure 3 and Figure 4. Although no effect was found for SOD in roots (Figure 3a), significant changes were observed in shoots (Figure 3c). In fact, when compared with CTL plants, SOD activity was inhibited by 45% upon single heat exposure, but a higher activity (28%) of this enzyme was recorded in response to the co-treatment compared to the CTL, with the ANOVA showing a significant interaction between both stress factors (Appendix A). CAT activity was similarly enhanced in roots of all stressed plants (Figure 3b) up to almost 100%, with a positive interaction being detected for this organ (Appendix A). Contrarily, in shoots, CAT was inhibited by 26% and 60% in response to single heat and co-exposure, respectively (Figure 3d), although no interaction was recorded (Appendix A).

APX activity (Figure 4a,d) was greatly enhanced in response to the combined action of the two stress factors in shoots (62%), but mainly in roots, where an increment of 129% in relation to the CTL was observed. The elevated activity of APX was also reported in roots upon individual exposure to heat (90%). Regarding DHAR activity (Figure 4b,e), compared to the CTL, it was noticeably enhanced only by the simultaneous exposure to the stressors (100% and 112% in roots and shoots, respectively). Indeed, the statistical analysis (Appendix A) shows that in both organs, there was a significant interaction between SALT and HEAT. Lastly, the individual salt stress inhibited GR by 25% in roots; however, when combined with heat, an increase of 31% was observed in relation to the CTL (Figure 4c), with the interaction of both conditions (salt and heat) being significant (Appendix A). In shoots (Figure 4f), the activity of this enzyme was elevated by 51% and 38% in plants under salt treatment and simultaneous exposure to both stress factors, respectively.

### 3.8. Principal Component Analysis (PCA)

To understand how different groups/conditions vary, and also to infer the correlation between all tested parameters, a PCA was carried out (Figure 5). The data obtained showed that the first component explained 52.48% and 62.29% of variance in roots and shoots, respectively, while the second accounted for 30.62% and 20.29%. Furthermore, it was observed that for roots (Figure 5a), SALT and CTL plants were almost grouped in the same quadrant (SALT in the fourth and CTL between the first and fourth), while HEAT and COMBINED plants were grouped separately in the second and third quadrants, respectively. In shoots (Figure 5b), although some proximity can be observed between SALT and COMBINED, the four treatments were distributed among the four quadrants (HEAT in the first, SALT in the second, COMBINED in the third, and CTL in the fourth), revealing that the dependent variables were affected differently by each experimental condition, as well as when comparing plant organs. It is also worth noting that more variables are related to CTL in shoots (namely, water content, CAT, K^+^, AsA, GSH, and H_2_O_2_) than in roots and that this group of plants is characterized by higher values of length, dry weight, and MDA in both organs. Interestingly, Na^+^ is associated with salinity treatments, especially COMBINED, in both organs and presents opposite relations to K^+^ and Ca^2+^ in roots. Lastly, in *S. lycopersicum* plants, the differences between the COMBINED and the remaining treatments concerning the redox status and lack of oxidative damage can be explained by the perceived negative correlation between MDA content and the general activation of the AOX system in roots, while in shoots, this was mostly observed for the enzymatic component of this system, along with proline and thiols.

## 4. Discussion

Currently, high temperatures and the salinization of soils and water are amongst the major environmental factors causing agricultural losses around the globe [43,44]. Despite these individual stresses having been extensively explored, little is known regarding their interaction, which frequently occurs simultaneously. Therefore, in this study, the response of tomato plants (*S. lycopersicum* var. *cerasiforme*) to the combination of heat and salinity was assessed in terms of growth and physiological performance to understand how plants cope and adjust their metabolism towards their co-occurrence.

### 4.1. The Combination of Heat and Salt Led to a Harsher Effect on Growth-Related Parameters

Here, plant growth, concerning root and stem elongation and biomass (Figure 1a,b,d,e), was impaired upon exposure to both salt and heat, but especially by the co-exposure treatment. Equivalent salt-induced declines in growth-related parameters have been reported in several crop plants [45,46,47,48], including tomato [49]. Such inhibitions are primarily correlated with a reduced water uptake, along with a negative interference in nutrient and ion ratios caused by the build-up of salts in the soil. Indeed, Na^+^ competes with K^+^ for transporters (AKT and HKT) due to their similarity in terms of ionic radius and hydration energy [50], resulting in an overaccumulation of Na^+^ and a lack of K^+^. Here, and corroborating previous studies on tomato [51,52], salt exposure, both individually and combined with heat, significantly enhanced the uptake of Na^+^. Once inside the plant, excessive salt becomes toxic as a result of the growing inability of cells to avoid the accumulation of Na^+^ and Cl^−^ ions in the cytoplasm and transpiration stream [10]. Besides K^+^, it is known that Ca^2+^ deficiency is often salt-induced, which may limit the efflux of Na^+^ to the apoplast via the Ca^2+^-dependent Salt Overly Sensitive (SOS) signaling pathway [10,50,53]. Once again, salt-exposed tomato plants exhibited lower levels of Ca^2+^. Additionally, and even though there is still a lot to unravel regarding the plants’ uptake of Mg [54], as well as the impacts of salinity on this process, it is generally expected to have a negative effect (Parihar et al. [10]). Curiously, the results presented herein show an opposite pattern, but a higher uptake of this nutrient might be related to its important role in plant growth, enzymatic activity, and photosynthesis—both as a key component of chlorophylls and as a vital player in CO_2_ fixation [55]—which are usually affected by Na^+^ toxicity [10]. Lastly, as the salt accumulation in soil hampers water uptake [47,48,56], by decreasing soil water potential, it is not surprising that these nutrient disbalances were also accompanied by a reduced water content (Figure 1f).

Similarly to the previous stressor, high temperatures significantly impaired tomato plants’ growth performance in both shoots and roots. Based on previous records, these heat-induced impacts are mostly due to disrupted water relations, damaged photosynthetic machinery, changes in membrane permeability, oxidative stress, and nutrient imbalance [9,44,57,58,59]. Nonetheless, and even though, in the present study, all ions analyzed (Na^+^, K^+^, Ca^2+^, and Mg^2+^) were altered upon heat exposure depending on the analyzed tissue, there appears to be no significant effect on nutrient uptake when looking at the whole plant—in accordance with the lack of macroscopic signs of nutrient deficiency. Indeed, the mechanisms by which high temperatures disturb nutrient uptake are still unclear and seem to be inconsistent [44,60,61]. Additionally, water content (Figure 1c,f) was unaffected by heat, similarly to other research on different tomato cultivars, namely that by Zhou et al. [14] and Rivero et al. [21], which is possibly related to the non-limiting irrigation. Nonetheless, it is important to take into account that, even with a possible enhancement in defense pathways and a generally unaffected nutrient uptake, heat-stressed plants still presented growth reduction, which might be related to heat-induced damage in the photosynthetic apparatus, leading to impaired carbon metabolism and reduced photoassimilate production [9,44].

Interestingly, when both stress factors were applied simultaneously, a stronger negative effect could be perceived on mineral absorption patterns (Table 1) and, consequently, on plant growth (Figure 1a,b,d,e,g), similarly to what was found in *Arabidopsis thaliana* (L.) Heynh, exposed to an identical combination of stressors [18]. Indeed, Ca^2+^ and K^+^ uptake was decreased in a harsher way than that found in SALT, and curiously, plants exposed to combined stressors also presented higher concentrations of Na^+^ than those in the single treatment. This may be a result of the ability of heat to reduce the activity of nutrient uptake proteins, most likely due to a lower root conductance or damage in enzymes, allowing for a greater influx of Na^+^ and a diminished uptake of Ca^2+^ and K^+^ [44,62]. This would limit the SOS pathway while also increasing the levels of Na^+^ in the cytosol, which presents a higher risk of toxicity and increased competition between Na^+^ and K^+^ for the binding sites of several key enzymes, culminating in a severe reduction in plant growth. However, the knowledge regarding the effects of heat stress on roots is limited, as is that regarding its impact on membrane transporters. Moreover, by impairing water uptake and leading to increased stomatal resistance, salinity could also have negatively influenced transpiration rate, an important cooling and nutrient distribution mechanism [10,63], increasing the plants’ susceptibility to heat stress. Although only a few records are available exploring the dynamics, in terms of physiological and biometrical impacts, of heat and salt co-exposure in *S. lycopersicum,* our data contrast with the reports of Rivero et al. [21] and Lopez-Delacalle et al. [20]. These authors, when exposing tomato plants cv. Optima to 120 mM NaCl for 72 h at 35 °C and cv. Boludo to 75 mM NaCl for 14 days at 35 °C, respectively, showed that the combination of both stressors prompted better growth, photosynthetic efficiency, and water and nutrient relations than those grown only under saline conditions. However, it is also important to consider that such contrasting results may arise from distinct tolerance thresholds between cultivars or varieties, as well as the employment of different experimental conditions that affect plant response and acclimation differently, namely growing plants in a soil-based system instead of hydroponics, as well as using a persistent or periodic exposure to high temperatures. Here, as the increased toxicity of Na^+^ may be affecting different processes—among them, water relations and photosynthesis (as supported by the lower concentrations of Mg^2+^)—growth might have been compromised due to the disruption of vital mechanisms, whether through a lack of resources or due to their allocation into defense pathways [e.g., accumulation of AOXs, osmolytes, and HSPs–, explaining the lack of macroscopic toxicity symptoms and oxidative damage (Section 4.2)], so that plant survival was ensured under these adverse conditions. In fact, and as reviewed by Margalha et al. [64], in conditions of disrupted nutrient uptake or ratios, the cross-talk between the two central nutrient-sensing kinases in plants leads to the induction of the one that ensures optimal nutrient allocation strategies and the inhibition of the one regulating nutrient use to promote cell growth and proliferation.

### 4.2. The Co-Exposure of Tomato Plants to Heat and Salinity, Individually or in Combination, Did Not Result in a Severe Oxidative Stress Condition

Even though the primary effects of salinity and heat are not related to oxidative stress, an excessive accumulation of ROS is fairly connected to a decline in growth and productivity in salt- [10] and heat- [44] exposed plants. However, in the present study, no major signs of ROS overaccumulation or membrane damage (measured as LP) were detected in plants subjected to either individual stressor, except in the roots of heat-stressed plants, where H_2_O_2_ levels were enhanced (Figure 2b). Nonetheless, the higher content of this ROS appears to be in equilibrium with the AOX capacity of tomato plants, as no oxidative damage, translated into LP, could be detected in this situation (Figure 2a).

Concerning the combined exposure, as in heat-exposed plants, a higher accumulation of H_2_O_2_ was found in roots, though O_2_^−^ content remained unaltered; additionally, in shoots, plants simultaneously subjected to salinity stress and high temperatures experienced a very noticeable decrease in this ROS in relation to all other experimental conditions (Figure 2a,b,d,e). In fact, the reduced O_2_^−^ content is in accordance with increased SOD activity—responsible for the dismutation of this ROS into H_2_O_2_ [12]. Thus, and while this would imply an increase in H_2_O_2_ content, the levels of this ROS were also reduced, possibly due to an efficient AOX response (as discussed in Section 4.3 and Section 4.4). Although, in some cases, reduced content of O_2_^−^ and/or H_2_O_2_ might be related to the production of other ROS, such as the hydroxyl radical (OH^.^), which is the main factor causing LP [12], no signs of oxidative damage could be found, namely in the MDA production (Figure 2c,f), suggesting that tomato plants are more likely investing in potent defense mechanisms to prevent salt- and/or heat-induced stresses.

### 4.3. The Simultaneous Effect of Heat and Salinity on Tomato Plants Results in Differential Activation Patterns of AOX Metabolites

Under water stress, which is often a consequence of salinity, heat, and drought exposure, plants tend to accumulate compatible organic solutes, such as proline [65]. In fact, proline is not only a powerful osmoprotectant but also a ROS scavenger—namely of OH. and singlet oxygen (^1^O_2_)—and a membrane stabilizer [12]. Thus, the exacerbated increase in the levels of this metabolite in plants exposed to salt, individually and in combination with heat (Table 2 and Table 3), may suggest a major role of proline in tomato’s tolerance response to this stressor. Such dramatic accumulation has already been documented in response to different concentrations of salt for several plant models [48,66,67,68,69,70], among them several distinct tomato varieties [71,72]. Indeed, proline acts on several fronts, including LP prevention, which probably explains the absence of membrane damage. Curiously, when both stressors were applied simultaneously, the levels of this osmoprotectant were noticeably enhanced in relation to CTL and HEAT treatments, but this increase was not as pronounced as in the SALT situation. Thus, and as already reported by Rivero et al. [21] and Lopez-Delacalle et al. [20], other defense pathways may be acting *in tandem* with proline in this response, as these authors report that decreased proline content (in comparison with the individual salt treatment) was accompanied by an increase in other osmolytes, such as glycine betaine. Additionally, considering not only the importance of proline accumulation but also the role of its catabolism in providing energy to the cell during stress conditions, particularly under situations of nutrient depletion (30 ATP equivalents are generated from the oxidation of one proline molecule [73]), it seems that proline metabolism plays an important role in the response of *S. lycopersicum* to the combined action of salt and heat stress. Nonetheless, the lower accumulation of this osmolyte—resulting from either its catabolism or reduced synthesis (associated with high carbon costs, around 10% of the plant weight [53])—can also be detrimental to these plants, as proline balances turgor pressure (affected by excess salt) and acts as a chaperone, preventing protein aggregation and denaturation as well as enzyme inhibition [73].

Knowing that no relevant symptoms of redox disorders were found upon the exposure to both stresses, either single or combined, the hypothesis of plant cells being able to ensure a proper redox state of proteins and other metabolites was raised. AsA, a powerful AOX involved in ROS scavenging via direct or indirect pathways [74], was not affected by either stress (individual or combined), in either its synthesis or its regeneration levels (Table 2 and Table 3). In fact, and although the opposite is frequently found [22,75,76], these results fall in accordance with the similar pattern of activity between the enzymes mediating AsA oxidation (APX) and its reduction (DHAR). Curiously, in heat-treated plants, especially in combination with salt, a slight tendency towards a higher oxidation of AsA was observed, this being in line with higher APX activity. However, in shoots, both salt-related treatments negatively influenced AsA accumulation, as has already been observed in *Brassica napus* L. [77] and *Vigna angularis* (Willd.) Ohwi and H. Ohashi [78] exposed to 100 mM NaCl. Nonetheless, as no signs of oxidative stress were detected and no major changes were found regarding the AsA redox status, it can be hypothesized that AsA biosynthesis might be only slightly downregulated, allocating energy and resources to other pathways. Indeed, even in the combined treatment, where APX activity was greatly enhanced, the lower AsA pool was still sufficient to ensure redox homeostasis of the cell, being accompanied by a great regeneration effort by DHAR.

The maintenance of reduced conditions within cells is of major importance in stressful conditions, with thiols (-SH) being excellent stress biomarkers [12]. The major non-protein thiol is GSH, also regarded as one of the main water-soluble AOXs [12]. Concerning salt stress, this metabolite appears to be more relevant in the aerial part of the plants, as no differences could be found in roots in either its content or its regeneration (Table 2). Nonetheless, in shoots, GSH levels severely decreased (Table 3) alongside a small decrease in total thiols and the increase in GR activity (Figure 4f) attempting to maintain the GSH pool. In fact, similar results were already found in several salt-stressed tomato cvs. (Gran brix and Marmande RAF) [79] and var. (*Super 2270*) [80]. This points towards a high oxidation rate, rather than degradation of this thiol, indicating that, since APX and DHAR activity remained unaltered (Figure 4a,b,d,e), GSH can act directly as a ROS scavenger or as a substrate for the ROS-scavenging function of glutathione peroxidase (GPX; EC 1.11.1.9), as H_2_O_2_ content was lower in this situation (SALT). Although the levels of GSSG were not quantified in order to confirm the mentioned hypothesis since the GSH/GSSG ratio is an important indicator of the cells’ redox status, the thorough analysis of the AsA-GSH cycle already provides a valid overview concerning GSH metabolism and the plants’ AOX potential.

When under heat stress, either individually or in combination with salt, the levels of total thiols increased in the roots of tomato plants due to a high accumulation of protein-bound thiols (Table 2). The main protein thiols are glutaredoxins and thioredoxins [81], with the latter having already been described as being important players in thermotolerance reactions [82,83] and thus contributing to the maintenance of the redox homeostasis. Aside from that, it is also important to note that in the combined treatment, GSH levels arose in roots (Table 2) alongside an overall upregulation of the AsA-GSH cycle (Table 2 and Figure 4a–c), highlighting its important role in ROS-scavenging reactions and in the maintenance of redox homeostasis. In shoots (Table 3), both temperature-related conditions presented a similar decrease in GSH content, although the principle behind that reduction can be different for both situations. Indeed, while the small decrease in GSH content in the heat treatment can be ascribed to a general irrelevance of the AsA-GSH cycle in this situation, coupled with a slight, but not significant, reduction in the activity of substrate-regenerating enzymes, the same did not occur in the combined treatment. Here, there was a clear induction of all enzymes pertaining to the AsA-GSH cycle (Figure 4), indicating a major role of this thiol in the proper functioning of this cycle, while also possibly acting by itself as a ROS scavenger or as a substrate for GPX. Actually, similar results have been reported when plants were exposed to high temperatures [84], even though a possible effect of heat stress on the biosynthesis of this thiol was suggested. However, in studies performed on tomato, heat, both as an individual stressor [85] and in combination with salt [20], led to an accumulation of this thiol, with or without the activation of the enzymatic cycle, granting more strength to the oxidation hypothesis than to that related to GSH degradation.

### 4.4. Combined Exposure to the Stressors Resulted in a Prompter Activation of the Enzymatic AOX Response, Especially the AsA-GSH Cycle Enzymes

Classified as the first line of defense, SOD catalyzes the detoxification of O_2_^−^ into H_2_O_2_ (Appendix A) [12]. Upon salt exposure, this enzyme was not activated in tomato plants, which is in accordance with the maintenance of the content of O_2_^−^ and H_2_O_2_ in roots. However, a slight tendency for salt-treated plants to present a higher SOD catalytic activity in shoots is also correlated with the small decrease in O_2_^−^ content that is herein reported. Considering the H_2_O_2_-scavenging enzymes (Appendix A), and even though several authors report the enhancement of various AOX enzymes in response to salt [86,87,88], only CAT was activated in the present work, and just in roots, explaining the maintenance of H_2_O_2_ in an organ that is commonly associated with salinity-induced oxidative stress, since it is the first contact point between the plant and the saline environment.

After exposure to high temperatures, O_2_^−^ levels remained unchanged (although a tendency to increase can be perceived), but a rise in the content of H_2_O_2_ was noticed in roots. This might be related to a SOD-mediated reaction, and while no changes in its activity were reported, high basal SOD levels could be sufficient to deal with mild stress conditions. Nonetheless, a tendency for SOD to possess higher activity in this treatment and organ can be noticed, as supported by Zhao et al. [22] and Liu and Huang [89]. Unsurprisingly, due to ROS accumulation, CAT and APX activities were enhanced in an effort to detoxify H_2_O_2_ and prevent oxidative damage. In fact, heat stress has already been shown to result in increased APX in roots [90], although for CAT, the opposite pattern is more common [89,90,91]. Moreover, APX activation did not result in an insufficient AsA pool, as both DHAR and GR remained unaffected, with efficient AsA regeneration possibly being ascribed to monodehydroascorbate (MDHAR, EC 1.6.5.4) action [12]. Contrarily to what was observed in roots, in shoots, no ROS were overaccumulated, and APX was not activated, while SOD and CAT were actually inhibited, effects also documented by Liu and Huang [89] and Djanaguiraman et al. [92] using 35 °C/25 °C and 40 °C/30 °C (day/night), respectively.

When plants were exposed to the combination of stressors, O_2_^−^ levels in roots were unaffected, which seems to agree with the maintenance of SOD activity also reported in this organ, an effect that opposes the activation of this AOX enzyme documented by Zhao et al. [22] in rice roots. Moreover, and though there was an activation of both H_2_O_2_-scavenging enzymes, APX and CAT, the AOX system did not fully detoxify this ROS, since H_2_O_2_ content was still higher than in CTL plants, but not at high enough levels to induce noticeable oxidative damage. In fact, is it possible that H_2_O_2_, under these still-higher concentrations, might serve as a signaling agent to prepare the plant for subsequent ROS bursts [93]. In shoots, and similarly to what Lopez-Delacalle et al. [20] reported, SOD was equally as activated as during the single salinity treatment, thus explaining the highly diminished levels of O_2_^−^. Nonetheless, an accumulation of H_2_O_2_ would be expected, which did not occur, possibly due to the enhanced activity of APX, a result also documented by Lopez-Delacalle et al. [20] in tomato plants, as well as by Koussevitzky et al. [94] and Zandalinas et al. [95] in *A. thaliana*, highlighting the importance of this enzyme in plants’ response to combined stress. Oppositely, CAT was inhibited in shoots; however, this might have little impact on the global AOX response due to its lower affinity to H_2_O_2_ [12] and the ability of these plants to maintain the redox homeostasis through other mechanisms. Overall, in plants exposed to the combination of salt and heat, the AOX enzymes, especially the ones involved in the AsA-GSH cycle, seem to be determinant to maintain redox homeostasis in shoots, while in roots, the enzymatic and the non-enzymatic components together play an important role in the response of tomato plants to the combined challenge of heat and salt stress.

## 5. Conclusions

Considering the data presented here, it is possible to assume that the AOX system, especially the AsA-GSH cycle, was of major importance in the response of *S. lycopersicum* L. plants to the co-exposure to heat and salt. Even so, the combination of these stressors not only resulted in higher impacts at both growth and biochemical levels, but also led to a higher accumulation of Na^+^ than the individual stresses. In fact, when observing the PCA (Figure 5), the combined treatment was plotted apart from all other treatments, with the main differences being associated with the higher accumulation of Na^+^ in both organs, which is paralleled by a decrease in the uptake of Ca^2+^ and K^+^, as well as a drastic reduction in plant growth. Thus, these results might suggest that, along with the accumulation of toxic ions, the already limited plant resources are being allocated towards defensive pathways to ensure survival under these adverse conditions, with the high carbon and energy costs associated with the stimulation of osmolytes and AOX enzymes possibly being another cause behind the severely reduced growth in co-treated plants. Further studies measuring the levels of ATP and NADPH are important to further verify this hypothesis.

As a follow-up message, subsequent research should be undertaken to complement what has been reported here (e.g., analysis of photosynthetic machinery and possible tolerance traits to these stressors), especially considering analyses at different time points and developmental stages, to achieve a robust insight into the overall effects of heat and salinity on crop physiology. Only then would it be possible to develop new and efficient ways to successfully alleviate the negative effects of these abiotic stresses, thus minimizing losses in crop productivity. Since tomato plants seemed to heavily invest in AOX mechanisms to counteract heat and salt co-exposure, the evaluation of AOX-promoting agents, such as biostimulants, phytohormones, or beneficial elements, could also represent a feasible tool to increase tomato tolerance to these two stressors.

## Figures and Tables

**Figure 1 antioxidants-11-00478-f001:**
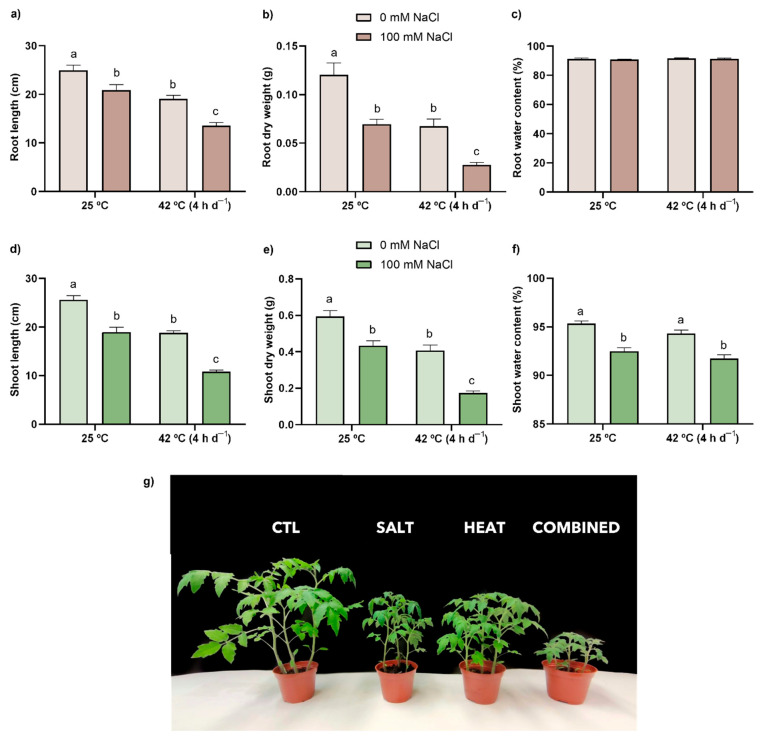
Length (**a**,**d**), dry weight (**b**,**e**), and water content (**c**,**f**) in roots (brown bars) and shoots (green bars), as well as the visual assessment (**g**) of tomato plants after a 21-day exposure to 42 °C (4 h d^−1^) and irrigation with (darker bars) or without (lighter bars) 100 mM NaCl. Values represent mean ± SEM (*n* ≥ 3). Significant differences (*p* ≤ 0.05) between treatments are indicated by different letters.

**Figure 2 antioxidants-11-00478-f002:**
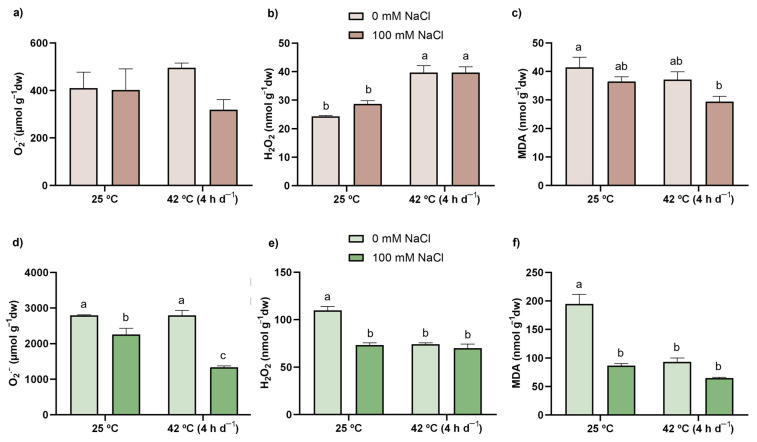
Levels of oxidative stress markers of tomato plants: O_2_^−^ (**a**,**d**), H_2_O_2_ (**b**,**e**) and MDA (**c**,**f**) content in roots (brown bars) and shoots (green bars) of tomato plants after a 21-day exposure to 42 °C (4 h d^−1^) and irrigation with (darker bars) or without (lighter bars) 100 mM NaCl. Values represent mean ± SEM (*n* ≥ 3). Significant differences (*p* ≤ 0.05) between treatments are indicated by different letters.

**Figure 3 antioxidants-11-00478-f003:**
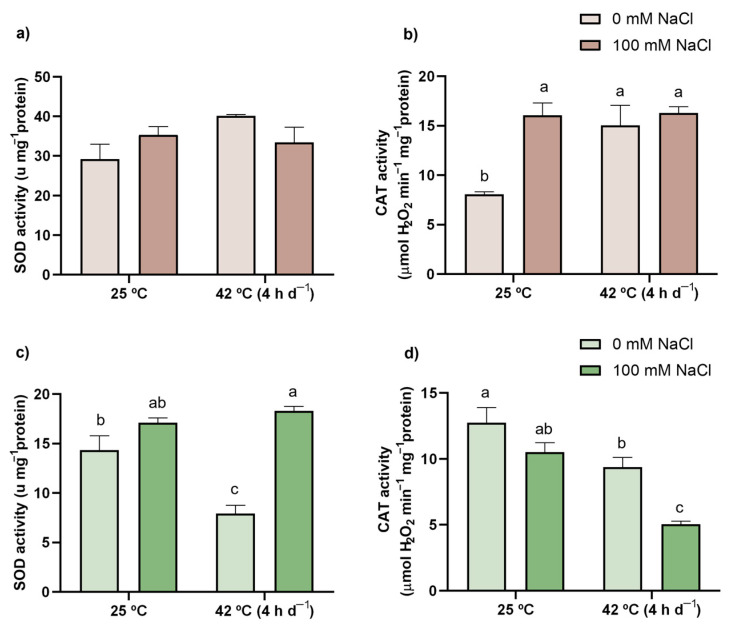
Activity levels of SOD (**a**,**c**) and CAT (**b**,**d**) in roots (brown bars) and shoots (green bars) of tomato plants after a 21-day exposure to 42 °C (4 h d^−1^) and irrigation with (darker bars) or without (lighter bars) 100 mM NaCl. Values represent mean ± SEM (*n* ≥ 3). Significant differences (*p* ≤ 0.05) between treatments are indicated by different letters.

**Figure 4 antioxidants-11-00478-f004:**
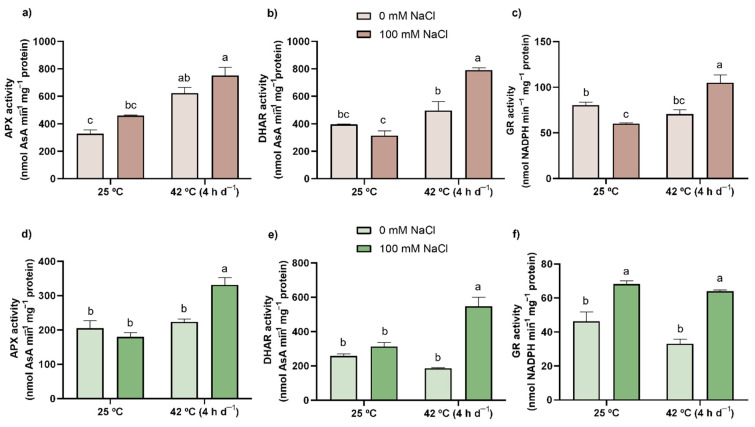
Activity levels of APX (**a**,**d**), DHAR (**b**,**e**), and GR (**c**,**f**) in roots (brown bars) and shoots (green bars) of tomato plants after a 21-day exposure to 42 °C (4 h d^−1^) and irrigation with (darker bars) or without (lighter bars) 100 mM NaCl. Values represent mean ± SEM (*n* ≥ 3). Significant differences (*p* ≤ 0.05) between treatments are indicated by different letters.

**Figure 5 antioxidants-11-00478-f005:**
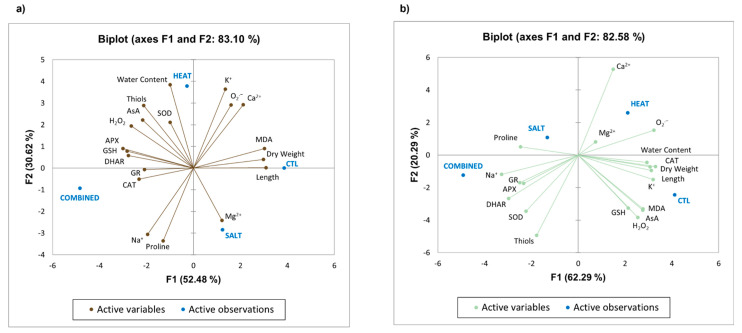
Biplot based PCA with first two principal components showing the differential response of roots (**a**) and shoots (**b**) of tomato plants to salt (irrigation with 100 mM NaCl), heat (exposure to 42 °C for 4 h per day), and combined stresses for 21 days.

**Table 1 antioxidants-11-00478-t001:** Effect of 21 days of salt (irrigation with 100 mM NaCl), heat (exposure to 42 °C for 4 h d^−1^), and combined stresses on the content of Na^+^, K^+^, Ca^2+^, and Mg^2+^ in roots and shoots of tomato plants. Values represent mean ± SEM (*n* ≥ 3). Significant differences (*p* ≤ 0.05) between treatments are indicated by different letters.

Parameter	CTL	SALT	HEAT	COMBINED
Root Na^+^ (mg g^−1^dw)	1.793 ± 0.003 ^d^	15.920 ± 0.021 ^b^	1.990 ± 0.026 ^c^	17.693 ± 0.015 ^a^
Shoot Na^+^ (mg g^−1^dw)	5.163 ± 0.5003 ^c^	25.380 ± 0.044 ^b^	4.180 ± 0.012 ^d^	40.050 ± 0.015 ^a^
Root K^+^ (mg g^−1^dw)	3.833 ± 0.019 ^b^	2.607 ± 0.015 ^c^	4.800 ± 0.012 ^a^	2.332 ± 0.002 ^d^
Shoot K^+^ (mg g^−1^dw)	11.583 ± 0.019 ^a^	8.050 ± 0.015 ^c^	9.917 ± 0.018 ^b^	7.570 ± 0.025 ^d^
Root Ca^2+^ (mg g^−1^dw)	0.487 ± 0.004 ^b^	0.323 ± 0.001 ^c^	0.557 ± 0.009 ^a^	0.181 ± 0.001 ^d^
Shoot Ca^2+^ (mg g^−1^dw)	2.000 ± 0.015 ^c^	2.343 ± 0.018 ^b^	2.757 ± 0.007 ^a^	1.767 ± 0.012 ^d^
Root Mg^2+^ (mg g^−1^dw)	2.547 ± 0.009 ^b^	3.193 ± 0.037 ^a^	2.527 ± 0.007 ^b^	2.397 ± 0.012 ^c^
Shoot Mg^2+^ (mg g^−1^dw)	6.097 ± 0.054 ^b^	6.737 ± 0.038 ^a^	5.730 ± 0.052 ^c^	5.473 ± 0.026 ^d^

**Table 2 antioxidants-11-00478-t002:** Effect of 21 days of salt (irrigation with 100 mM NaCl), heat (exposure to 42 °C for 4 h d^−1^), and combined stresses on the content of proline, AsA (total, AsA, DHA, and AsA/DHA), GSH, and thiols (total and protein/non-protein) in roots of tomato plants. Values represent mean ± SEM (*n* ≥ 3). Significant differences (*p* ≤ 0.05) between treatments are indicated by different letters.

Parameter (Roots)	CTL	SALT	HEAT	COMBINED
Proline (mg g^−1^dw)	0.099 ± 0.02 ^c^	5.820 ± 0.114 ^a^	0.088 ± 0.039 ^c^	4.320 ± 0.357 ^b^
Total AsA (µg g^−1^dw)	7.273 ± 0.500	7.803 ± 0.444	8.607 ± 0.406	8.240 ± 0.633
AsA (µg g^−1^dw)	1.980 ± 0.665	1.867 ± 0.044	1.933 ± 0.079	1.960 ± 0.269
DHA (µg g^−1^dw)	5.917 ± 0.173	6.023 ± 0.3868	6.723 ± 0.3480	6.280 ± 0.6201
AsA/DHA	0.338 ± 0.047	0.297 ± 0.012	0.280 ± 0.012	0.263 ± 0.019
GSH (nmol g^−1^dw)	252.5 ± 13.5 ^b^	233.2 ± 1.95 ^b^	295.0 ± 18.2 ^ab^	374.5 ± 36.4 ^a^
Total thiols (µmol g^−1^dw)	1.306 ± 0.023 ^b^	1.116 ± 0.038 ^b^	1.785 ± 0.111 ^a^	1.739 ± 0.032 ^a^
Non-protein/Protein thiols	0.232 ± 0.045	0.300 ± 0.050	0.179 ± 0.018	0.146 ± 0.021

**Table 3 antioxidants-11-00478-t003:** Effect of 21 days of salt (irrigation with 100 mM NaCl), heat (exposure to 42 °C for 4 h d^−1^), and combined stresses on the content of proline, AsA (total, AsA, DHA, and AsA/DHA), GSH, and thiols (total and protein/non-protein) in shoots of tomato plants. Values represent mean ± SEM (*n* ≥ 3). Significant differences (*p* ≤ 0.05) between treatments are indicated by different letters.

Parameter (Shoots)	CTL	SALT	HEAT	COMBINED
Proline (mg g^−1^dw)	1.112 ± 0.154 ^c^	29.930 ± 2.265 ^a^	0.593 ± 0.023 c	18.540 ± 1.117 ^b^
Total AsA (µg g^−1^dw)	18.84 ± 0.93 ^a^	13.20 ± 1.65 ^b^	14.19 ± 1.03 ^ab^	12.86 ± 0.47 ^b^
AsA (µg g^−1^dw)	6.497 ± 0.224	5.340 ± 0.932	4.173 ± 0.376	4.397 ± 0.095
DHA (µg g^−1^dw)	12.34 ± 0.80 ^a^	7.87 ± 0.73 ^b^	10.01 ± 0.67 ^ab^	8.46 ± 0.38 ^b^
AsA/DHA	0.53 ± 0.03 ^ab^	0.67 ± 0.06 ^a^	0.42 ± 0.02 ^b^	0.52 ± 0.01 ^ab^
GSH (nmol g^−1^dw)	1039.0 ± 9.8 ^a^	733.1 ± 18.6 ^c^	904.6 ± 41.6 ^b^	870.0 ± 28.2 ^b^
Total thiols (µmol g^−1^dw)	7.862 ± 0.720 ^a^	7.292 ± 0.350 ^ab^	5.324 ± 0.438 ^b^	8.809 ± 0.198 ^a^
Non-protein/Protein thiols	0.110 ± 0.004 ^b^	0.094 ± 0.009 ^b^	0.194 ± 0.007 ^a^	0.113 ± 0.006 ^b^

## Data Availability

Data is contained within the article or Appendix A.

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
