# Peer review of "Impact of Combined Heat and Salt Stresses on Tomato Plants—Insights into Nutrient Uptake and Redox Homeostasis"

_antioxidants, 2022, doi:10.3390/antiox11030478_

Round 1
Reviewer 1 Report
In this revised manuscript, the authors addressed most of my comments. However, they never responded to one of my major points described below.
They measured all physiological and biochemical parameters at one time point of stress application, namely 21-days after stress treatment, which weakens the main conclusion of the manuscript. What was the rationale to choose 21-days for the measurement of these parameters? Since the physiological and biochemical parameters measured in this study can be changed depending on the duration of the stress treatment, it is important to measure these parameters at least two or three time points of stress treatment.
Reviewer 2 Report
The article analyzes the separate and combined effects of salinity and high temperature on the physiological and biochemical characteristics of young tomato plants. It was shown that a long-term (21 days) combined action of stressors led to a stronger damaging effect on plants. It is quite obvious that prolonged exposure to two different stressors will result in a stronger impact. However, the authors emphasize that the absorbed ions as a result of NaCl treatment have a great influence on the state of the antioxidant system. Perhaps this gives some novelty to the study.
The article is written very carefully, but often too detailed. However, before making a conclusion about the possibility of its publication, I would like the authors to correct some points.
- The title of the article is not entirely correct. It is not clear what is meant by Disbalance? How can there be a Disbalance between ion uptake and homeostasis of the redox system? What do the authors understand by the term "Redox homeostasis"?
- The last phrase in the Abstract is unclear.
- The text of the manuscript is written in very complicated phrases that are difficult to understand. It is advisable to change the style of writing an article, making the phrases shorter and easier to read.
- The Introduction contains a lot of excess and unnecessary information. The basis of the work is not clear (2nd paragraph of the Introduction). If authors tried to prove that salinity and high temperature affected plants simultaneously, then drought must also be included, which more often proceed jointly with heat. The last phrase contains an incorrect statement that the threshold of heat stress of most crops being around 25-35 °C (line 69-70).
- Sections 2.1 and 2.2 contain conflicting information about the number of vessels with plants for individual trials and the total number of vessels.
- Section 2.3 states that the dry weight was determined after air drying at 60 °C. However, this is not a dry weight, which is determined by drying at 100-105 °C. It is air-dry weight.
- In the Results section, it is clearly seen that heat and salinity impacts predominantly reduced or did not change markers of oxidative stress. Therefore, it is impossible to speak about disturbances of the oxidative status under the action of stressors on tomato plants. At the same time, it is not clear how this could be.
- The discussion is very long and needs to be shortened and streamlined.
Round 2
Reviewer 1 Report
I have no further comments.
Reviewer 2 Report
The presented new version of the article has been revised in accordance with the comments in the review. All comments were taken into account by the authors, and the corresponding changes were made to the manuscript. The only remark is that the authors still kept long phrases and wordy comments in the Discussion. I believe that the article needs some editorial processing.
This manuscript is a resubmission of an earlier submission. The following is a list of the peer review reports and author responses from that submission.
Round 1
Reviewer 1 Report
The manuscript sent for review seems interesting and worth publishing, however, it contains significant editorial errors. The authors have included some of the results in the Discussion, which is incorrect. From Figure 4, the order of the graphs should be changed, that is: first, the changes within each cultivar, and then the differences between the cultivars in the different phases of the experiment. Materials and methods should indicate the percentage of moisture in the top soil layer after adding a drying layer. In Conclusions it should be added that betaine is not an important osmotically active substance in the process of alfalfa frost resistance.
The English language seems to be correct, although I would suggest reviewing the text again to correct some minor mistakes.
In conclusion, I suggest that the authors correct the errors and send the text again.
Reviewer 2 Report
Dear authors,
it is interesting paper about important crop being studied in the context of unfavorable conditions that may interfere with its productivity.
I appreciate that combined stress conditions (very likely occuring in Mediterranean region) were used in the present study, since the ecosystems are being affected by multiple stresses that interact. Although, response of plant systems can be different under single vs. combined stress treatments, as reported also in this study.
In fact, I do not have any questions or comments and recommend the paper to be published in the present form.
Reviewer 3 Report
This is a very nice paper that is focused on a highly important subject (stress combination). The authors did a good job in conducting the experiments and interpreting the results. However, several points need to be addressed before the manuscript could be accepted:
- The authors found a negative effect of the stress combination (salt and heat) on tomato. However, they failed to mention earlier work showing a negative effect of this stress combination on Arabidopsis:
Suzuki N, Bassil E, Hamilton JS, Inupakutika MA, Zandalinas SI, Tripathy D, Luo Y, Dion E, Fukui G, Kumazaki A, Nakano R, Rivero RM, Verbeck GF, Azad RK, Blumwald E, Mittler R. ABA Is Required for Plant Acclimation to a Combination of Salt and Heat Stress. PLoS One. 2016 Jan 29;11(1):e0147625. doi: 10.1371/journal.pone.0147625. PMID: 26824246; PMCID: PMC4733103.
- The authors did not measure the ratio of GSH/GSSG that is one of the most important parameters of redox in cells. They need to highlight this deficiency in their study and discuss it.
- Since the authors found that APX is important, they should mention earlier work about the importance of APX to stress combination:
Koussevitzky S, Suzuki N, Huntington S, Armijo L, Sha W, Cortes D, Shulaev V, Mittler R. Ascorbate peroxidase 1 plays a key role in the response of Arabidopsis thaliana to stress combination. J Biol Chem. 2008 Dec 5;283(49):34197-203. doi: 10.1074/jbc.M806337200. Epub 2008 Oct 13. PMID: 18852264; PMCID: PMC2590703.
Zandalinas SI, Balfagón D, Arbona V, Gómez-Cadenas A, Inupakutika MA, Mittler R. ABA is required for the accumulation of APX1 and MBF1c during a combination of water deficit and heat stress. J Exp Bot. 2016 Oct;67(18):5381-5390. doi: 10.1093/jxb/erw299. Epub 2016 Aug 6. PMID: 27497287; PMCID: PMC5049388.
- The authors suggest in the abstract that the energetic cost of detoxifying ROS and synthesizing osmo- protectants caused a reduction in growth. This is likely correct, but they need to propose ways of measuring this. Maybe propose to measure NADPH and ATP levels in future studies.
Reviewer 4 Report
The authors analyzed the effects of combined salt and heat stresses on the physiological and biochemical response of tomato plants. By measuring several parameters, including the contents of Na+, K+, Ca2+, and Mg2+, ROS levels, and antioxidant metabolites and several AOX enzymes, they concluded that the accumulation of toxic ions, coupled with the high energy costs from the stimulation of osmolyte synthesis and AOX enzymes, heavily compromised the ability of plants to grow properly during the combined salinity and heat stresses. Although the manuscript contains some interesting information regarding the effects of combined salt and heat stresses on tomato seedling growth, the data are preliminary, and additional experiments should be done to improve the manuscript.
Major points;
- They evaluated the effects of combined salt and heat stresses on the growth of tomato seedlings but never assessed their effects on fruit development, ripening, or yield. If the effects of single and combined stresses on fruit development, ripening, or yield are measured, it will greatly improve the manuscript.
- They measured all physiological and biochemical parameters at one time point of stress application, namely 21-days after stress treatment, which weakens the main conclusion of the manuscript. What was the rationale to choose 21-days for the measurement of these parameters? Since the physiological and biochemical parameters measured in this study can be changed depending on the duration of the stress treatment, it is important to measure these parameters at least two or three time points of stress treatment.
- The pictures showing the phenotypes of tomato plants under single and combined stress conditions should be added.
Minor points;
- It is described in the bottom part of the Abstract that “accumulation of toxic ions, coupled with the high energy costs (ATP and reducing power) from the stimulation of osmolyte synthesis and AOX enzymes”. Where are the data supporting high energy costs? The levels of ATP and reducing power were never measured.
- Although the manuscript is well written overall, English grammar should be corrected in some parts of the manuscript.